# Heart Rate Variability in Hyperthyroidism: A Systematic Review and Meta-Analysis

**DOI:** 10.3390/ijerph19063606

**Published:** 2022-03-18

**Authors:** Valentin Brusseau, Igor Tauveron, Reza Bagheri, Ukadike Chris Ugbolue, Valentin Magnon, Jean-Baptiste Bouillon-Minois, Valentin Navel, Frédéric Dutheil

**Affiliations:** 1Endocrinology Diabetology and Metabolic Diseases, CHU Clermont-Ferrand, University Hospital of Clermont-Ferrand, F-63000 Clermont-Ferrand, France; itauveron@chu-clermontferrand.fr; 2Institut Génétique, Reproduction & Développement (iGReD), CNRS, INSERM, University of Clermont Auvergne, F-63000 Clermont-Ferrand, France; 3Department of Exercise Physiology, University of Isfahan, Isfahan 81746-73441, Iran; will.fivb@yahoo.com; 4Institute for Clinical Exercise & Health Science, School of Health and Life Sciences, University of the West of Scotland, Glasgow G1 1XW, UK; u.ugbolue@uws.ac.uk; 5Department of Biomedical Engineering, University of Strathclyde, Glasgow G1 1XW, UK; 6Physiological and Psychosocial Stress, CHU Clermont-Ferrand, University of Clermont Auvergne, F-63000 Clermont-Ferrand, France; valentinmagnon@hotmail.fr (V.M.); j-baptiste.bouillon-minois@uca.fr (J.-B.B.-M.); fred_dutheil@yahoo.fr (F.D.); 7Emergency Department, CHU Clermont-Ferrand, University of Clermont Auvergne, F-63000 Clermont-Ferrand, France; 8Translational Approach to Epithelial Injury and Repair, GreD, CNRS, INSERM, University of Clermont Auvergne, F-63000 Clermont-Ferrand, France; valentin.navel@hotmail.fr; 9Ophthalmology, CHU Clermont-Ferrand, University Hospital of Clermont-Ferrand, F-63000 Clermont-Ferrand, France; 10Occupational and Environmental Medicine, CHU Clermont-Ferrand, University Hospital of Clermont-Ferrand, F-63000 Clermont-Ferrand, France

**Keywords:** thyroid, biomarker, autonomic nervous activity, prevention, public health

## Abstract

Objective: Cardiovascular effects of thyroid hormones may be measured through heart rate variability (HRV). We sought to determine the impact of hyperthyroidism on HRV. Design: A systematic review and meta-analysis on the impact of hyperthyroidism on HRV. Methods: PubMed, Cochrane, Embase and Google Scholar were searched until 20 August 2021 for articles reporting HRV parameters in untreated hyperthyroidism and healthy controls. Random-effects meta-analysis was stratified by degree of hyperthyroidism for each HRV parameter: RR intervals (or Normal-to-Normal intervals—NN), SDNN (standard deviation of RR intervals), RMSSD (square root of the mean difference of successive RR intervals), pNN50 (percentage of RR intervals with >50 ms of variation), total power (TP), LFnu (low-frequency normalized unit) and HFnu (high-frequency), VLF (very low-frequency), and LF/HF ratio. Results: We included 22 studies with 10,811 patients: 1002 with hyperthyroidism and 9809 healthy controls. There was a decrease in RR (effect size = −4.63, 95% CI −5.7 to −3.56), SDNN (−6.07, −7.42 to −4.71), RMSSD (−1.52, −2.18 to −0.87), pNN50 (−1.36, −1.83 to −0.88), TP (−2.05, −2.87 to −1.24), HFnu (−3.51, −4.76 to −2.26), and VLF power (−2.65, −3.74 to −1.55), and an increase in LFnu (2.66, 1.55 to 3.78) and LF/HF ratio (1.75, 1.02 to 2.48) (*p* < 0.01). Most parameters had ES that was twice as high in overt compared to subclinical hyperthyroidism. Increased peripheral thyroid hormones and decreased TSH levels were associated with lower RR intervals. Conclusions: Hyperthyroidism is associated with a decreased HRV, which may be explained by the deleterious effect of thyroid hormones and TSH. The increased sympathetic and decreased parasympathetic activity may have clinical implications.

## 1. Introduction

The thyroid gland and the autonomic nervous system are closely linked by their control center, the hypothalamus, and by their effects on the cardiovascular system [1,2]. Hyperthyroidism is a common global health problem and a risk factor for cardiovascular mortality [3]. One of the main complications of hyperthyroidism is cardiac arrhythmias, most often supraventricular, and may be caused by sympathovagal imbalance. Indeed, the clinical manifestations of hyperthyroidism (tachycardia, palpitation, systolic arterial hypertension) suggest β-adrenergic stimulation and dysautonomia [4,5,6,7]. Dysautonomia means a change in the function of the autonomic nervous system can negatively affect the health of a person [8]. Sympathovagal imbalance is associated with an increased risk of ventricular arrhythmias and cardiac mortality [9,10], which can be measured by the study of heart rate variability (HRV). HRV is the variation between two consecutive heartbeats related to the continuous interaction between the two arms of the autonomic nervous system, sympathetic and parasympathetic [11]. HRV is a sensitive, quantitative and non-invasive tool for the study of autonomic nerve function [12,13,14]. High HRV suggests an adaptable and dynamic autonomic nervous system [15]. Low HRV is a marker of cardiovascular risk and represents an abnormal or restricted ability of the autonomic nervous system to maintain homeostasis [16,17]. Although the evaluation of HRV in hyperthyroidism has been assessed in several studies, conflicting results have been reported. Indeed, the degree and type of autonomic imbalance and its contribution to cardiovascular abnormalities in hyperthyroidism are not fully understood [18]. These conflicting results may be partly explained by variation in patient demographic profiles and differences in disease type, severity and duration. Many studies have shown a tendency for HRV depression with an impaired cholinergic reserve, providing a logical explanation for the increased sympathetic activity in hyperthyroidism. If these results are reproducible, it may contribute to the understanding of the susceptibility to cardiac arrhythmias in hyperthyroidism and indicate possible early therapeutic intervention. In addition, there is no consensus on the decreased levels of HRV parameters in hyperthyroidism. Two biochemical entities are distinguished: overt hyperthyroidism, with a prevalence of 0.5% of the general population [19], and subclinical hyperthyroidism, with 1.8% [20]. Few studies have comprehensively evaluated the role of the most common variables, such as age, sex, body mass index (BMI), blood pressure or biochemical thyroid function on HRV parameters [21,22].

Therefore, we aimed to conduct a systematic review and meta-analysis of the impact of untreated overt or subclinical hyperthyroidism on HRV parameters. A secondary objective was to identify the most frequently reported explanatory variables.

## 2. Methods

### 2.1. Literature Search

We reviewed all studies measuring HRV in patients with untreated hyperthyroidism and healthy controls. We searched the main article databases (PubMed, Cochrane Library, Embase and Google Scholar) with the following keywords: (“hyperthyroidism” OR “hyperthyroid”) AND (“heart rate variability” OR “HRV”) until 20 August 2021. All articles compatible with our inclusions criteria were included, independently of article language and years of publication. To be included, studies had to describe our main primary outcome i.e., the measurement of HRV parameters in untreated hyperthyroid patients and healthy controls. We imposed no limitation on the regional origin or the nature of the control group. We excluded studies that assessed the effects of treated hyperthyroidism in adults on HRV parameters, animal studies, studies in children, conferences, congresses, seminars and studies without frequency or time domains for HRV parameters or without controls. Studies needed to be primary research. In addition, reference lists from all publications meeting the inclusion criteria were manually searched to identify any further studies that were not found with the electronic search. Ancestry searches were also completed on previous reviews to locate other potentially eligible primary studies. The search strategy is presented in Figure 1 and Appendix A. Two authors (VB and RB) conducted the literature searches, reviewed the abstracts and articles independently, checked suitability for inclusion, and extracted the data. When necessary, disagreements were solved with a third author (FD).

### 2.2. Data Extraction

The primary endpoint was the analysis of HRV parameters in untreated hyperthyroid patients and in healthy controls. Traditionally, HRV is measured by linear methods [12], and most studies dealing HRV and dysthyroidism used linear HRV measurement methods. In the time domain, we analyzed RR intervals (or normal-to-normal intervals—NNs), standard deviation of RR intervals (SDNN), percentage of adjacent NN intervals differing by more than 50 ms (pNN50) and the square root of the mean squared difference of successive RR intervals (RMSSD). The time domain of HRV can be decomposed into its frequency components by the spectral analysis technique, either with the fast Fourier transform algorithm or with autoregressive modeling [12]. This is analogous to a prism that refracts light into its wavelength components [12]. In the spectral domain, we analyzed the total power (TP), low frequency (LF, 0.04 ± 0.15 Hz), high frequency (HF, 0.15 ± 0.4 Hz) and very low frequency (VLF, 0.003 ± 0.04 Hz), and the LF/HF ratio. Power is the energy found in a frequency band [23]. LF and HF powers are absolute powers, reported in units of ms^2^ (square milliseconds). LFnu and HFnu are normalized powers, called relative powers, in the LF and HF bands, comprising a derived index that is calculated by dividing LF or HF by an appropriate denominator representing the relevant total power: LFnu = LF/(LF + HF) and HFnu = HF/(LF + HF). These normalized powers allow direct comparison of the frequency domain measurements of two patients despite a large variation in specific band power and total power [24]. LF power represents both sympathetic and parasympathetic activity and is associated with SDNN, but LFnu emphasizes the control and balance of cardiac sympathetic behavior [12]. HF power and HFnu represent the most efferent parasympathetic activity [25] and are associated with RMSSD and pNN50 [23]. As for SDNN, both sympathetic and parasympathetic activities contribute to VLF with uncertainty about the physiological mechanisms responsible for activity in this band [26]. The LF/HF ratio is the most sensitive indicator of sympathovagal balance [12], which was also calculated and reported in this meta-analysis. Secondary outcomes included hyperthyroidism characteristics (duration and etiology of hyperthyroidism, free thyroxine—fT4, free triiodothyronine—fT3, thyroid-stimulating hormone—TSH), clinical parameters (BMI, blood pressure, treatments, other diseases), electrical measures such as heart rate, and sociodemographic parameters (age, sex, smoking).

### 2.3. Quality of Assessment

We used the Scottish Intercollegiate Guidelines Network (SIGN) criteria to check the quality of included articles with the dedicated evaluation grids. For clinical trials, checklists consist of 10 items if randomized and 7 items if non-randomized, based on the main causes of bias [27]. We also used the SIGN score for cohort and cross-sectional studies, in two sections: design of the study (14 items), and overall evaluation (3 items). There were 4 possibilities of answers (yes, no, can’t say or not applicable) (Appendix A). We also used the “STrengthening the Reporting of OBservational studies in Epidemiology” (STROBE—32 items/sub-items) for cohort and cross-sectional studies [28] and the Consolidated Standards of Reporting Trials (CONSORT—37 items/subitems) for randomized trials [29]. We attributed one point per item or sub-item, to achieve a maximal score of 32 or 37, respectively, then converted this into a percentage.

### 2.4. Statistical Considerations

We used Stata software (v16, StataCorp, College Station, TX, USA) for the statistical analysis [30,31,32,33,34]. Main characteristics were synthetized for each study population and reported as mean ± standard deviation (SD) for continuous variables and number (%) for categorical variables. When data could be pooled, we conducted random effects meta-analyses (DerSimonian and Laird approach) for each HRV parameter comparing patients with untreated hyperthyroidism with healthy controls [35]. A negative effect size (ES, standardized mean differences—SMD) [36] denoted lower HRV in patients than in controls. An ES is a unitless measure, centered at zero if the HRV parameter did not differ between hyperthyroidism patients and controls. An ES of −0.8 reflects a large effect i.e., a large HRV decrease in patients compared to controls, −0.5 a moderate effect, and −0.2 a small effect. Then, we conducted meta-analyses stratified on biochemical status of hyperthyroidism, subclinical or overt. We evaluated heterogeneity in the study results by examining forest plots, confidence intervals (CI) and I-squared (I^2^). I^2^ is the most common metric to measure heterogeneity between studies, ranging from 0 to 100%. Heterogeneity is considered low for I^2^ < 25%, modest for 25 < I^2^ < 50%, and high for I^2^ > 50%. We also searched for potential publication bias by examining funnel plots of these meta-analyses. We verified the strength of our results by conducting further meta-analyses after exclusion of studies that were not evenly distributed around the base of the funnel. When possible (sufficient sample size), meta-regressions were proposed to study the relationship between each HRV parameter, and clinically relevant parameters (age, sex, blood pressure, BMI), hyperthyroidism status (subclinical or overt) and biological relevant parameters (fT3, fT4, TSH). Results are expressed as regression coefficients and 95% CI. *p*-Values less than 0.05 were considered statistically significant.

## 3. Results

An initial search produced a possible 638 articles (Figure 1). Removal of duplicates and use of the selection criteria reduced the number of articles reporting the evaluation of HRV in untreated hyperthyroidism to 22 articles [37,38,39,40,41,42,43,44,45,46,47,48,49,50,51,52,53,54,55,56,57,58]. All included articles were written in English.

Among the 22 studies included, 15 were cross-sectional [41,42,44,45,46,47,48,49,50,53,54,55,56,57,58], five were prospective [37,38,39,40,51], one was retrospective [52] and one was a randomly controlled trial (RCT) [43]. Included studies were published from 1996 to 2019 and conducted across four continents (Europe—11 studies, Asia—eight studies, America—two studies, Africa—one study). All included articles aimed to compare HRV between patients with untreated hyperthyroidism and controls without hyperthyroidism [37,38,39,40,41,42,43,44,45,46,47,48,49,50,51,52,53,54,55,56,57,58].

Sample sizes ranged from 20 [38,50,54] to 8759 [52], for a total of 10,811 patients: 1002 with untreated hyperthyroidism and 9809 healthy controls.

Thyroid function was described clinically and biologically in all studies. Twelve studies included overt hyperthyroidism [37,38,39,40,41,42,47,48,49,54,56,58], six subclinical [44,45,50,52,55,57], and four both [43,46,51,53]. Most studies included newly diagnosed and untreated hyperthyroid patients before initiation of the treatment [37,38,39,40,41,42,47,48,49,50,54,56,57].

Recording of HRV measurements was ambulatory, spontaneous breathing with normal daily activity in all studies. Most studies used ECG, achieved in a resting supine position, to determine HRV [37,38,43,46,47,48,49,52,55,56,57] between four [48] and 30 min [40,41,42], except eight studies using a 24 h Holter-ECG [39,44,45,50,51,53,54,58]. Parameters reported were both time and frequency domains in most studies, except five studies that reported only time domain [41,43,44,47,51] and one only frequency domain [55].

More details on study characteristics (Appendix A), aims and quality of articles, inclusion and exclusion criteria, characteristics of population, characteristics of hyperthyroidism, and HRV measurements and analysis are described in Appendix A.

### 3.1. Meta-Analyses of HRV Values in Untreated Hyperthyroidism

The main results of the meta-analysis are presented in Figure 2. Compared with healthy controls, we noted strong evidence (*p* < 0.001) that hyperthyroid patients had significantly lower RR intervals (ES = −4.63, 95% CI −5.70 to −3.56), SDNN (−6.07, −7.42 to −4.71), RMSSD (−1.52, −2.18 to −0.87), pNN50 (−1.36, −1.83 to −0.88), TP (−2.05, −2.87 to −1.24), LF power (−1.18, −1.82 to −0.54), HF power (−1.75, −2.51 to −0.99), HFnu (−3.51, −4.76 to −2.26) and VLF power (−2.65, −3.74 to −1.55), and higher LFnu (2.66, 1.55 to 3.78) and LF/HF ratio (1.75, 1.02 to 2.48) (Appendix A).

### 3.2. Meta-Analysis Stratified by Subclinical or Overt Status

In comparison to healthy controls, the following HRV parameters were decreased in both overt hyperthyroidism and in subclinical hyperthyroidism, respectively: RR-intervals (ES = −6.97, 95% CI −9.07 to −4.88, and −0.98, −1.79 to −0.16), SDNN (−7.87, −10.08 to −5.67, and −3.45, −5.28 to −1.62), pNN50 (−1.64, −2.48 to −0.8, and −1.12, −1.86 to −0.37), LF power (−1.32, −2.27 to −0.37, and −0.69, −1.31 to −0.07), HF power (−1.99, −3.12 to −0.86, and −0.96, −1.83 to −0.09), and VLF power (−2.27, −3.47 to −0.73, and −1.33, −1.96 to −0.71), whereas LFnu increased (3.19, 1.86 to 4.53, and 0.59, 0.02 to 1.17). Most of the aforementioned parameters had ES that was twice as high in overt compared to subclinical hyperthyroidism. Some HRV parameters were only modified in overt hyperthyroidism: lower RMSSD (−1.87, −2.92 to −0.82), TP (−2.05, −2.87 to −1.24) and HFnu (−4.24, −5.75 to −2.74), and higher LF/HF ratio (1.75, 1.02 to 2.48), while those parameters did not differ in subclinical hyperthyroidism (Appendix A). All meta-analyses had a high degree of heterogeneity (I^2^ > 80%), except for parameters explored by a few studies in subclinical hyperthyroidism (LFnu, HFnu, VLF).

### 3.3. Meta-Regressions and Sensitivity Analyses

An increase in fT3 and fT4 was associated with lower RR intervals (coefficient = −0.47, 95% CI −0.71 to −0.22 and −0.10, −0.19 to −0.01, respectively); while an increase in TSH was associated with higher RR intervals (35.7, 2.53 to 68.9). In addition, patients with overt hyperthyroidism had lower RR intervals (−6.00, −9.75 to −2.25) than those with subclinical hyperthyroidism. Age was associated with higher RR intervals (0.35, 0.13 to 0.58), TP (0.35, 0.15 to 0.56) and HFnu (1.13, 0.35 to 1.91), and lower LFnu (−0.75, −1.38 to −0.12) and LF/HF ratio (−0.30, −0.55 to −0.05). BMI was associated with higher RR intervals (1.69, 0.03 to 3.35), TP (0.41, 0.07 to 0.76), LF power (0.53, 0.19 to 0.86), HF power (0.72, 0.32 to 1.11) and VLF power (2.70, 1.73 to 3.68), and lower LF/HF ratio (−0.69, −1.3 to −0.08). An increase in systolic blood pressure was associated with a lower RMSSD (−0.30, −0.50 to −0.10) (Figure 3 and Appendix A).

The meta-analyses were rerun after excluding studies that were not evenly distributed around the base of the funnel (Appendix A) and showed similar results (data not shown).

## 4. Discussion

The main results showed a decreased HRV in patients with hyperthyroidism, which may be explained by the deleterious effect of thyroid hormones and TSH. The increased sympathetic and decreased parasympathetic activity may have clinical implications. Some other factors, such as age or BMI, should also be considered in a clinical perspective.

### 4.1. Deleterious Effects of Thyroid Hyperfunction on HRV

The cardiovascular effects of thyroid hormones occur either directly through nuclear receptors [4] or indirectly by the sympathoadrenergic system [59]. Excess thyroid hormones has a direct chronotropic effect on the sinus node [60,61]. Changes in HRV are not only related to chronotropic effects. For example, propanolol is one of the most effective treatments for heart rate and did not alter HRV parameters [62]. Hyperthyroidism is characterized by a hyperkinetic state, similar to that induced by catecholamine excess [6], but serum and urine catecholamine levels are normal or decreased in hyperthyroidism [63,64]. The increased density and sensitivity of β-adrenergic receptors to catecholamines in hyperthyroidism may explain the increase in sympathetic activity [65,66,67]. More specifically, we showed a sympathovagal imbalance in hyperthyroidism. Vagal inhibition was more intense than increased sympathetic activity, with a greater decrease in HF power than LF power. As expected, TP decreased markedly (cardiac vagal control) as HF is its main contributor—two-thirds—whereas LF and VLF contribute one-third [12,68]. HRV is decreased mainly because of a large decrease in vagal activity [12,68]. Then, RR intervals decreased in patients with subclinical hyperthyroidism, and further decreased in overt hyperthyroidism. Moreover, an increase in fT3 and fT4, and a decrease in TSH, were related to a decrease in RR intervals. HRV parameters may indirectly reflect the severity of hyperthyroidism [58]. Subclinical hyperthyroidism appears to be an intermediate cardiovascular state between euthyroidism and overt hyperthyroidism, a continuum related to thyroid hormone excess [46,52]. An increase in sympathetic activity seems to be the first modification of the sympathovagal balance, which may be due to the decrease in TSH [44,45,50,52,55,57]. However, these results should be treated with caution because those studies only reported some selected HRV parameters. This parasympathetic inhibition may be due to the action of thyroid hormones on centers regulating autonomic functions [69,70] and on cardiac M2-muscarinic receptors [66], and increased adrenergic reactivity may be due to the main effects of abnormal TSH concentrations [53].

### 4.2. Clinical Implications

Decreased vagal tone and increased sympathetic activity in hyperthyroidism have important clinical implications. Thyroid hormones play a role in arrhythmogenesis with a risk of atrial fibrillation [71], which may be related to decreased HRV [72]. For example, a high incidence of supraventricular arrhythmias has been reported in overt hyperthyroidism women with very low HRV [58]. Increased sympathetic modulation and vagal inhibition were observed before the onset of paroxysmal atrial fibrillation [73], which may explain the increased prevalence of atrial fibrillation in these patients. A decreased HRV should strengthen the idea of treating subclinical hyperthyroidism [74]. However, early antithyroid therapy remains contradictory [44,74]. Indeed, if antithyroid treatment allows reversibility of HRV abnormalities, it would constitute an additional argument to treat subclinical hyperthyroidism in order to avoid rhythmic complications in these patients. Patients with decreased vagal tone are more susceptible to cardiovascular disease [75,76] with increased cardiac morbidity and mortality without apparent heart muscle damage [49]. It has also been shown that decreased TP predicts an increased risk of sudden cardiac death [77] and total cardiac mortality [78], that decreased LF is a strong predictor of sudden death independently of other variables [74], and that decreased VLF is an indicator of increased cardiac mortality in patients after myocardial infarction [79,80]. These data suggest that HRV parameters may be a marker of increased mortality in hyperthyroid patients. Physical activity and hyperthyroidism have the same effects on HRV, i.e., a concomitant sympathetic activation and decreased vagal tone [81]. Hence, many hyperthyroid patients are intolerant to exercise due to a reduced ability to increase cardiac output [82,83], in addition to the usual musculoskeletal manifestations of hyperthyroidism [84].

### 4.3. Other Variables Related to HRV in Hyperthyroidism

Age was associated with higher RR, TP, HFnu, and lower LFnu and LF/HF ratio. Thus, age was linked with an increased HRV in hyperthyroidism. However, in the general population, older age is associated with a decrease in HRV [85,86] due to decreased parasympathetic regulation [87]. Our results may be explained by the fact that, in our meta-analysis, younger patients had more severe hyperthyroidism and a high prevalence of Graves’ disease [88]. We demonstrated that an increase in systolic blood pressure was associated with lower RMSSD, i.e., a decrease in parasympathetic activity. No study has previously evaluated this relationship in hyperthyroidism. Conflicting results have been reported in the general population, with elevated blood pressure associated with either an increase [89] or a decrease [90] in HRV. It has also been suggested that decreased autonomic nerve function precedes the development of clinical hypertension [91]. We also demonstrated that increased BMI was associated with higher RR intervals, TP, LF, HF and VLF power, and lower LF/HF ratio, i.e., increasing HRV with increased parasympathetic activity. However, an increase in BMI is associated with lower HRV [92,93]. Hyperthyroid patients often presented a weight loss, resulting in a significantly lower mean BMI than healthy controls. In malnourished subjects, there is a decrease in HFnu with an increase in LFnu and LF/HF ratio [94]; hence, normalization of BMI may improve HRV. BMI does not distinguish between lean and fat tissue [95,96]. Interestingly, HRV may be more related to body composition than to BMI, and especially to body fat [97,98], which is lowered in hyperthyroid patients [99].

### 4.4. Limitations

All meta-analyses have limitations, including those of the individual studies from which data are obtained, and are theoretically subject to publication bias [100]. Although our meta-analysis was based on a moderate number of studies, the use of broader keywords in the search strategy limited the number of missing studies [101]. The quality of the studies varied despite our rigorous criteria for selecting studies in the meta-analysis [43,57,58]. Indeed, most studies were cross-sectional, and only a single RCT was included [43], precluding robust conclusions for our meta-analyses [101]. Data collection and inclusion/exclusion criteria, although similar, were not identical in each study, which may have affected our results [102]. In addition, all studies except one [52] were monocentric, limiting the generalizability of our results [102]. We limited the influence of extreme results and heterogeneity by exclusion of outliers [103,104]. Moreover, declarative data from studies are a source of putative bias [100]. Studies also differed in measurement conditions, such as in the duration of recording of HRV parameters [48,58]. We did not undertake meta-analysis of non-linear assessments of HRV, but non-linear assessment has been poorly studied in hyperthyroidism and is controversial; its results are non-proportional, maximizing minimal or major changes [105,106]. The etiology and duration of hyperthyroidism were poorly reported, precluding further analysis. Similarly, the lack of data on spectral analysis of subclinical hyperthyroidism did not allow conclusions to be drawn on the type and degree of sympathovagal imbalance.

## 5. Conclusions

HRV is markedly decreased in hyperthyroid patients. Increased sympathetic and decreased parasympathetic activity may be explained by the deleterious cardiovascular effects of thyroid hormones. The benefits of HRV assessment in the evaluation and monitoring of the severity of hyperthyroidism should be further investigated, given its potential as a noninvasive, reliable, and pain-free measurement.

## Figures and Tables

**Figure 1 ijerph-19-03606-f001:**
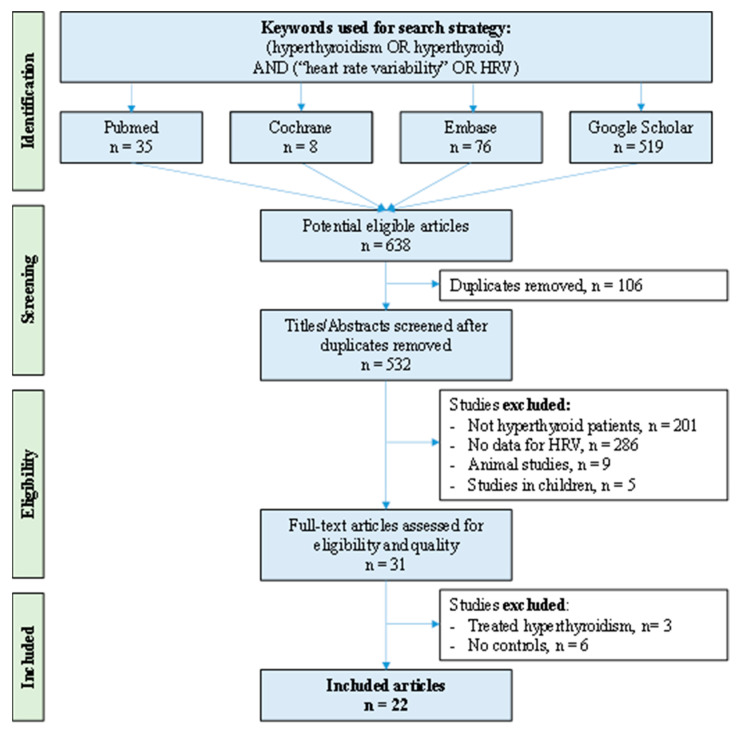
Flow chart. We followed the Preferred Reporting Items for Systematic Reviews and Meta-Analyses (PRISMA) guidelines for the search strategy. HRV: Heart rate variability.

**Figure 2 ijerph-19-03606-f002:**
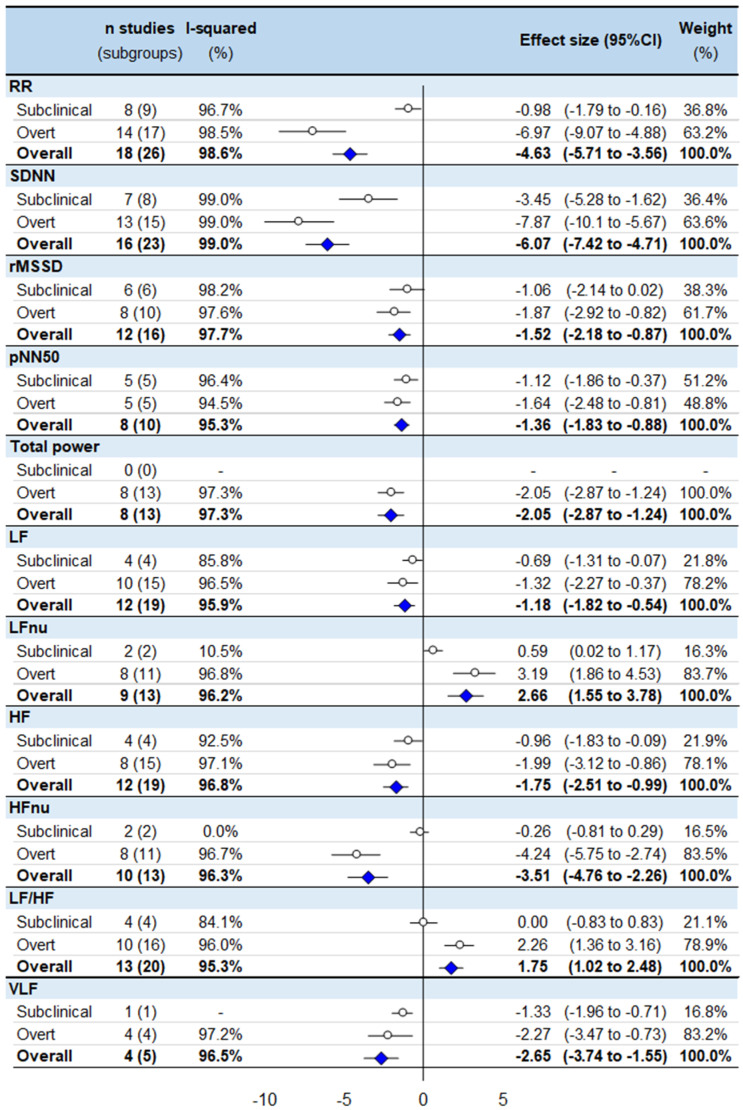
Meta-analysis of HRV parameters of untreated hyperthyroid patients compared with controls. RR: RR intervals (or normal-to-normal intervals—NNs), SDNN: standard deviation of RR intervals, pNN50: percentage of adjacent NN intervals differing by more than 50 ms, RMSSD: the square root of the mean squared difference of successive RR intervals, LF: low frequency, LFnu: low frequency-normalized units, HF: high frequency, HFnu: high frequency-normalized units, LF/HF ratio: low frequency/high frequency ratio, VLF: very low frequency.

**Figure 3 ijerph-19-03606-f003:**
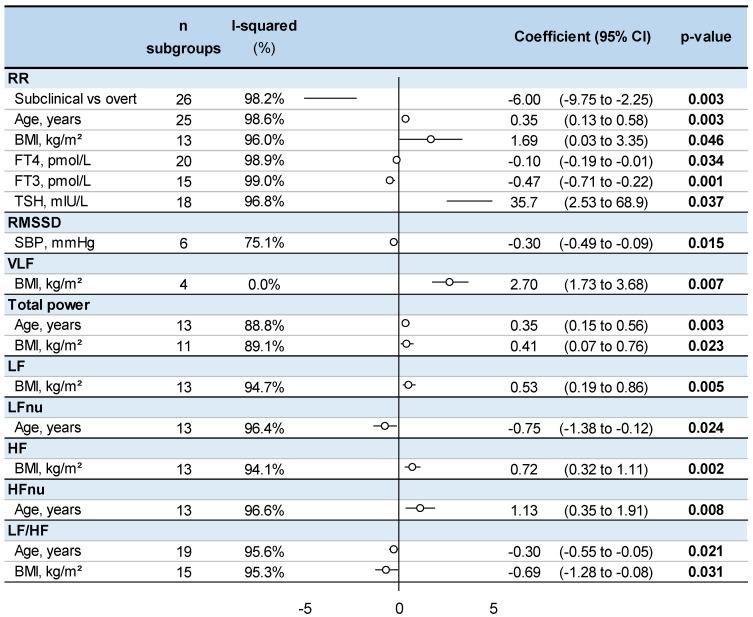
Meta-regressions of significant factors influencing heart rate variability in untreated hyperthyroid patients (exhaustive meta-regressions are presented in Appendix A). RR: RR intervals (or normal-to-normal intervals—NNs), BMI: body mass index, FT4: free thyroxine, FT3: free triiodothyronine, TSH: thyroid-stimulating hormone, RMSSD: the square root of the mean squared difference of successive RR intervals, SBP: systolic blood pressure, VLF: very low frequency, LF: low frequency, LFnu: low frequency-normalized units, HF: high frequency, HFnu: high frequency-normalized units, LF/HF ratio: low frequency/high frequency ratio.

## Data Availability

All relevant data were included in the paper.

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
