# Peer review of "Heart Rate Variability in Hyperthyroidism: A Systematic Review and Meta-Analysis"

_ijerph, 2022, doi:10.3390/ijerph19063606_

Round 1
Reviewer 1 Report
The authors present a study examining the the impact of hyperthyroidism on heart rate variability. However, as the authors say, etiology and duration of hyperthyroidism were poorly reported, which makes a deeper analysis difficult.
A few of minor points:
- Table 1 could be attached as complementary material and include a summary of the characteristics of the works at the beginning of the results
- Figure 1 is very helpful in understanding the review process – thank you for including this.
Author Response
Veuillez, s'il vous plait, consulter la pièce jointe.

Reviewer 2 Report
This study evaluated the Heart rate variability in hyperthyroidism using the meta-analysis. The authors found that HRV is markedly decreased in hyperthyroid patients.
The analysis of systematic review quite well described. I have no more comments on this article.
As this is the review article, I cannot say the originality of manuscript in that this is the summation of the previous articles.
The review process of systematic review well described, and the authors used funnel plot to find the effects of negative publication.
The conclusion follows the initial assumption of the hypothesis of manuscript.
And the data of HRV and hyperthyroidism is well described in figure 2.
Additionally, the authors described well the result of secondary analysis in the results section with subheading.
Additionally, the authors present primary and secondary purpose of this study well in the introduction.
Thus, I think this article should be published.
Reviewer 3 Report
The topic of thyroid hormone impact on HR and the risk of arrythmias is widely known, however this systematic review and metanalysis covers an extensive review of the literature what is it strong side.
In general is well prepared paper. The only suggestion for the authors, if they could, in the discussion be a bit more specifically about the potential translation of their results into clinical practice especially in case of subclinical hyperthyroidism in which the indication to the treatment is still discussed.
